# CNN-LSTM with Geometric Transformation and Clinical Knowledge Integration for No-Reflow and Slow Flow Prediction from Intravascular Ultrasound Images

**Keishi Kaneko**[1]                                             KANEKO.KEISHI@CHIBA-U.JP

[1] *Department of Medical Engineering, Graduate School of Science and Engineering, Chiba University, Chiba, Japan*

**Takeshi Nishi**[2]                                             TAKESHI24@HOTMAIL.CO.JP

[2] *Department of Cardiology, Beacon Kalamazoo Hospital, 1521 Gull Road, Kalamazoo, MI 49048, USA*

**Yukihiro Nomura**[3]                                           YNOMURA@CHIBA-U.JP
**Toshiya Nakaguchi**[3†]                                        NAKAGUCHI@FACULTY.CHIBA-U.JP

[3] *Center for Frontier Medical Engineering, Chiba University, Chiba, Japan*

## Abstract

During percutaneous coronary intervention (PCI), embolized plaque can cause complications such as no-reflow and slow flow phenomena. Predicting these complications is clinically crucial but challenging due to a complex interplay of various factors, such as plaque composition and vessel diameter. While intravascular ultrasound (IVUS) provides valuable lesion information, models using only IVUS images as input yield limited predictive performance. In this study, we propose an approach that integrates geometric transformation and clinical knowledge into a deep learning pipeline to improve complication prediction. Specifically, we apply a polar coordinate transformation to IVUS images to improve spatial feature extraction efficiency and eliminate non-region-of-interest artifacts such as the IVUS catheter. Furthermore, we compute four statistics of the attenuation angle within the plaque across multiple frames—mean, maximum, minimum, and standard deviation—and integrate them into a convolutional neural network–long short-term memory (CNN-LSTM) architecture. Experimental results on a dataset of 312 cases demonstrate that our proposed integration of polar transformation and clinical features improves the area under the curve (AUC) from 0.62 to 0.66, indicating its potential to enable more accurate quantitative risk assessment.

**Keywords:** Intravascular Ultrasound, Attenuated Plaque, CNN-LSTM

## 1. Introduction

Percutaneous coronary intervention (PCI) is a standard treatment for coronary artery disease; however, plaque embolization can cause complications such as the no-reflow phenomenon and slow flow, which occur postoperatively in approximately 10–20% of cases (Babapoor et al., 2022). These complications, once they occur, require post-procedural treatment such as vasodilators, which carries the risk of serious adverse effects including acute hypotension and arrhythmias. Therefore, establishing preventive strategies to avoid these complications before they occur, rather than managing them after onset, is of critical

---

†. Corresponding author

importance. In current clinical practice, distal protection devices are effective preventive measures for high-risk lesions; however, their routine application to all cases is impractical due to the burden of device preparation, making accurate pre-procedural risk stratification essential. However, intravascular ultrasound (IVUS) interpretation requires specialized expertise, and accurate prediction by visual assessment alone is challenging because multiple factors—such as vessel diameter and plaque composition—interact in a complex manner, necessitating methods that quantitatively assess complication risk from IVUS images. Our preliminary study used an approach in which IVUS images were input directly into a ResNet-34-long short-term memory (LSTM) model and achieved an area under the curve (AUC)—a threshold-independent metric—of only 0.62, indicating that predictive performance requires improvement. A significant association between attenuated plaque and no-reflow has been reported (Wu et al., 2011), but their scoring system is manual, motivating an automated and objective approach. In this study, we propose a method that explicitly integrates attenuation angle statistics computed from physician annotations into a convolutional neural network–LSTM (CNN-LSTM) architecture.

## 2. Methods

### 2.1. Dataset

The IVUS data used in this study were obtained from IVUS videos acquired at Chiba University Hospital. For each case, 13 frames were extracted at 1 mm intervals, centered on the frame corresponding to the plaque lesion, with six frames on either side. The dataset comprises 312 cases: 39 with no-reflow, 35 with slow flow, and 238 without complications. No-reflow and slow flow cases are combined as the positive class. Each frame is annotated with the attenuation angle direction by a physician experienced in IVUS image interpretation. All experiments were evaluated using 5-fold cross-validation, and results are reported as the mean across folds.

### 2.2. Proposed Method

First, a polar coordinate transformation is applied to each IVUS frame. The transformation origin is set to the image center and resamples the image to $224 \times 254$ pixels using bicubic interpolation. The leftmost 30 pixels, corresponding to the IVUS catheter region, are then removed, yielding a final input size of $224 \times 224$ pixels. This preprocessing eliminates non-region-of-interest artifacts and unfolds the circular vessel cross-section into a rectangular representation, improving spatial feature extraction efficiency. Next, the attenuation angle annotations are used to compute four statistics across the 13 frames—mean, maximum, minimum, and standard deviation—forming the attenuated plaque feature vector. The model consists of ResNet-34 and an LSTM; the attenuated plaque feature vector is concatenated to the LSTM output and passed through a fully connected layer to predict the probability of complication occurrence. We also evaluate a comparison method using a Region of Interest (ROI) mask of the plaque region concatenated to the IVUS image along the channel dimension, where physician annotations are used for annotated frames and the segmentation model of Nishi et al. (2021) fine-tuned on annotated frames of this dataset is used for frames in which severe attenuation makes annotation infeasible.

## 3. Results

Table 1 presents the ablation study results, including comparisons with the scoring-based approach of Wu et al. (2011) (AUC: 0.65) and a single physician's prediction made by reviewing IVUS images from the same dataset (AUC: 0.56). The proposed method (AUC: 0.66) outperformed Wu et al. (2011)'s approach, indicating that image features extracted from IVUS by the CNN provide additional predictive value beyond the attenuation angle alone. All AI models also outperformed the physician in AUC. While polar transformation alone did not improve AUC over the baseline (0.62), integrating attenuated plaque features alone raised it to 0.63. The proposed method combining both achieved the highest AUC of 0.66, with reduced variance ($\pm 0.07$) compared with the baseline ($\pm 0.12$), suggesting greater stability. In contrast, adding the ROI mask to the proposed method yielded an AUC of only 0.63, indicating that the ROI mask does not necessarily contribute to improved predictive performance.

Table 1: Ablation study results.

| Method | Polar Transform | Attenuated Plaque Features | Accuracy | Precision | Recall | F1 | AUC |
|---|---|---|---|---|---|---|---|
| Wu et al. (2011) | — | — | $0.58 \pm 0.16$ | $0.31 \pm 0.06$ | $0.54 \pm 0.30$ | $0.36 \pm 0.10$ | $0.65 \pm 0.08$ |
| Physician prediction | — | — | $0.56 \pm 0.07$ | $0.32 \pm 0.05$ | $\mathbf{0.72} \pm 0.08$ | $\mathbf{0.44} \pm 0.06$ | $0.65 \pm 0.09$ |
| ResNet-34-LSTM | ✗ | ✗ | $0.54 \pm 0.12$ | $0.31 \pm 0.08$ | $0.71 \pm 0.18$ | $0.43 \pm 0.09$ | $0.62 \pm 0.12$ |
| ResNet-34-LSTM | ✓ | ✗ | $0.64 \pm 0.13$ | $\mathbf{0.36} \pm 0.14$ | $0.44 \pm 0.09$ | $0.38 \pm 0.11$ | $0.62 \pm 0.12$ |
| ResNet-34-LSTM | ✗ | ✓ | $0.56 \pm 0.11$ | $0.31 \pm 0.07$ | $0.64 \pm 0.20$ | $0.40 \pm 0.05$ | $0.63 \pm 0.09$ |
| ResNet-34-LSTM (Ours) | ✓ | ✓ | $0.60 \pm 0.11$ | $0.35 \pm 0.10$ | $0.62 \pm 0.11$ | $0.43 \pm 0.05$ | $\mathbf{0.66} \pm 0.07$ |
| ResNet-34-LSTM (+ROI Mask) | ✓ | ✓ | $\mathbf{0.65} \pm 0.08$ | $0.35 \pm 0.08$ | $0.53 \pm 0.22$ | $0.41 \pm 0.09$ | $0.63 \pm 0.11$ |

## 4. Conclusion

In this study, we proposed a CNN-LSTM pipeline integrating polar coordinate transformation and attenuated plaque features for predicting PCI complications from IVUS images. The ablation study demonstrated that combining both components improved the AUC from the baseline of 0.62 to 0.66, outperforming physician prediction in AUC (AUC: 0.56). These results indicate that explicitly integrating geometric transformation of IVUS images with clinical knowledge is effective in improving the predictive performance of deep learning models for complication prediction. The failure of the ROI mask to improve predictive performance is likely attributable to insufficient accuracy of plaque region segmentation. Future work includes further improving predictive performance through the integration of other lesion features, such as calcified plaque, as well as clinical information.

## Acknowledgments

This study was approved by the Ethics Committee of the Graduate School of Medicine, Chiba University (Approval No. 3270).

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
