# OpenReview forum: "CNN-LSTM with Geometric Transformation and Clinical Knowledge Integration for No-Reflow and Slow Flow Prediction from Intravascular Ultrasound Images"
_MIDL.io/2026/Short_Papers — MIDL 2026 - Short Papers Poster_

### Official Review · Reviewer_Zzo9 · 2026-05-03
**Nice approach, with good preliminary experiments, but unclear improvement**

**Rating:** 3
**Confidence:** 4

**Review:**

The paper is clearly written, with a sound method and careful experiment, including an ablation study. It is very unclear though if the proposed polar conversion and expert input are truly an improvement. Recall, in particular drastically drops when these additions are made over a classical ResNet-34-LSTM network.

**Summary:**

The paper presents an approach to predict complications in PCI procedures, based on IVUS images and expert-based measurements of the attenuation angle. Experiments are performed on a dataset of 312 cases, 238 without complications. They show a definite improvement over Wu et al's previous work, but the proposed addition to the ResNet-34-LSTM backbone do not bring a clear improvement (base recall of 0.71 against 0.62 with the proposed version).

**Strengths:**

- The paper addresses a clinically important topic
- The method is interesting in that it combines automatic image analysis, using a deep neural network, with clinical measures performed by an expert
- Experiments are perform on real data, using 5-fold validation, with an ablation study
- Results improve over previous work

**Weaknesses:**

- Results appear somewhat inconclusive in Table 1. The ResNet-34-LSTM alone seems to be an equally good candidate, questioning the actual impact of using polar coordinates and expert input
- 5-fold validation is used but no dispersion measures are provided with the results, making it difficult to assess the significance of the quite small differences between the ablated versions of the proposed approach

**Justification Of Rating:**

Nice approach, with good preliminary experiments, but unclear improvement.

---

### Decision · Program_Chairs · 2026-05-08

Accept (Poster)